# RT-Remover: A Real-Time Video Object Removal by Composing Tracking and Removal in Auto-Regressive Diffusion Transformers

## Abstract

With the rapid advancement of video diffusion, video editing techniques, especially video object removal, have garnered increasing attention. Existing methods generally rely on separate object tracking and inpainting stages, leading to complex and slow pipelines that are unsuitable for real-time and interactive applications. This paper is committed to designing a real-time video object remover with the minimum latency, termed as **RT-Remover**. To this end, we introduce three key innovations in this paper to enable the real-time object removal in videos. First, different from previous methods that perform tracking and inpainting individually, we compose them into a joint process. Our model only requires an initial mask for the first frame from the user and automatically removes the target objects across the whole video. Second, we leverage an auto-regressive diffusion model for a real-time video object remover. We use an auto-regressive form to predict the next chunk based on previous chunks, while use diffusion model to iteratively predict the current chunk. Meanwhile, we incorporate a fixed-length key-value cache to minimize both memory usage and computational overhead. Third, to further speed up the inference, we propose to distill the auto-regressive diffusion model using distribution matching distillation and flow matching loss, and thus reduce the number of sampling steps from 25 to 2 while preserving background consistency. All three contributions significantly simplify the pipeline and enable real-time performance. Our method achieves *33 FPS* and *0.12s latency* on a 5090 GPU with our trained faster VAE. Extensive experiments show that our approach achieves the lowest latency among existing methods while maintaining competitive visual quality.

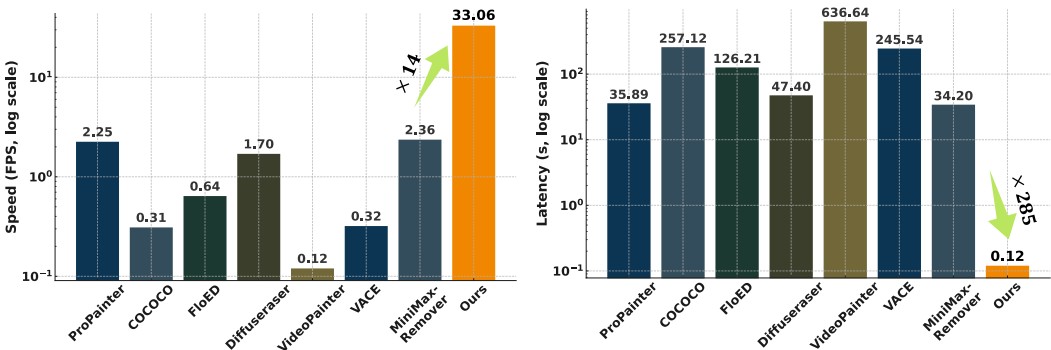

(a) The Inference Speed of Different Methods.    (b) The Inference Latency of Different Methods.

Figure 1: The inference efficiency were obtained on a 5090 GPU with an input resolution of $480 \times 832$ over 81 frames. The inference speed is calculated as *Num Frames/Inference Duration*. Latency is defined as the time elapsed from receiving an input frame to producing the corresponding output frame. *RT-Remover reduces inference time and latency by a factor of 14 and 285, respectively, compared to Minimax-Remover.*

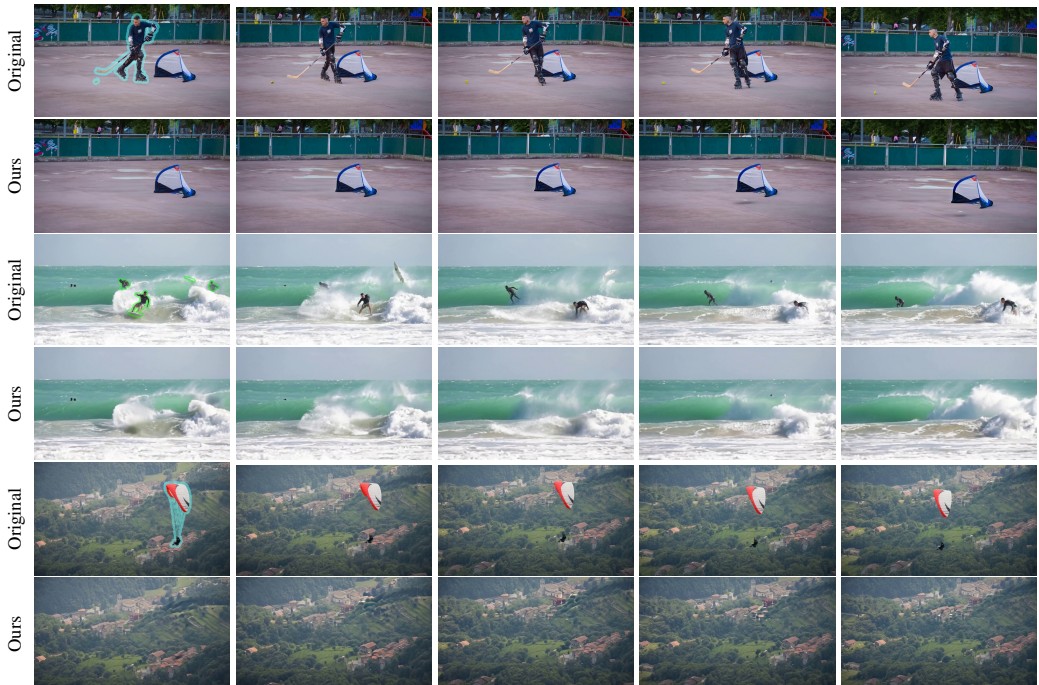

Figure 2: Visualization of our RT-Remover. We present three sets of results. For instance, the first row shows some frames from the source video, and the second row displays the outcomes after object removal.

# 1 INTRODUCTION

In recent years, video editing techniques have witnessed rapid advancements. Early methods such as AnimateDiff (Guo et al., 2023), VideoComposer (Wang et al., 2024), and VideoCrafter2 (Chen et al., 2024) leveraged convolutional neural networks to train diffusion models. With access to millions of training videos, these approaches achieved impressive visual quality. More recently, diffusion transformer (DiT) architectures have become dominant in video generation (Peebles & Xie, 2022). Models like HunyuanVideo (Kong et al., 2024) and Wan2.1 (Wang et al., 2025) are trained on billions of videos, producing not only visually appealing outputs but also demonstrating strong motion consistency.

Concurrently, video editing methods have evolved alongside video generation techniques. Broadly, video editing can be divided into two categories: (1) **General-purpose editing**, and (2) **Specific-purpose editing**. General-purpose editing aims to manipulate videos based on user-provided prompts, allowing flexible modifications across any region or type of edit. Representative methods include Tune-A-Video (Wu et al., 2023), TokenFlow (Geyer et al., 2023), and Senorita (Zi et al., 2025b). In contrast, specific-purpose editing targets a defined task with a specialized design, often receiving tailored inputs and producing constrained outputs. Examples include video stylization and video inpainting. Generally, specific-purpose methods exhibit greater reliability and stability within their targeted domains, which has contributed to their growing popularity in recent years.

Among various specific-purpose tasks, video object removal has gained significant attention. A common approach combines an object tracker with a video inpainting model: the tracker generates a mask sequence, while the inpainting model fills the masked regions with plausible content. Research on video inpainting has a long history. For instance, ProPainter (Zhou et al., 2023) is a non-diffusion method that uses flow completion to guide a transformer-based inpainting module. More recent techniques employ diffusion models. FFF-VDI (Lee et al., 2025) incorporates optical flow to guide the inpainting process. Senorita-Remover (Zi et al., 2025b) uses dual contrastive objectives to learn object removal and content generation simultaneously. FloED (Gu et al., 2024) employs a dual-branch network and a two-stage training strategy for high-quality results. DiffuEraser (Li et al., 2025) leverages DDIM inversion and vision priors to enhance inpainting fidelity. Notably, MiniMax-

Table 1: Comparison of latency improvements from Traditional Remover (i.e., Minimax-Remover) to RT-Remover, highlighting the sources of acceleration.

| Method | Mask Tracking | VAE | Encode Mask | Modeling Type | Sampling Steps | Latency (s) | Speedup Ratio |
|---|---|---|---|---|---|---|---|
| Traditional Remover | ✓ | Wan2.1 VAE | ✓ | Diffusion | 6 | 34.20 | 1.0 |
| Model-1 | ✗ | Wan2.1-VAE | ✓ | Diffusion | 6 | 15.16 (**-19.04**) | 2.25 |
| Model-2 | ✗ | Wan2.1-LeanVAE | ✓ | Diffusion | 6 | 8.79 (**-6.37**) | 3.89 |
| Model-3 | ✗ | Wan2.1-LeanVAE | ✗ | Diffusion | 6 | 8.64 (**-0.15**) | 3.95 |
| Model-4 | ✗ | Wan2.1-LeanVAE | ✗ | AR+Diffusion | 6 | 0.32 (**-8.32**) | 106.87 |
| RT-Remover (ours) | ✗ | Wan2.1-LeanVAE | ✗ | AR+Diffusion | 2 | 0.12 (**-0.20**) | **285.00** |

Remover (Zi et al., 2025a) introduces adversarial training to improve removal quality, achieving appealing results with only six sampling steps without using classifier-free guidance.

However, current object removal methods still suffer from many drawbacks, the most notable one is the slow inference and high latency. This drawback is mainly due to three perspectives. First, current methods require a separate object tracking step prior to removal to get the mask sequence, which adds significant preprocessing time. Second, existing methods are non-auto-regressive and require simultaneous processing of all frames in a clip, resulting in high computational latency. Third, current approaches often depend on multiple sampling steps, further increasing inference time.

To address the above challenges, we propose **RT-Remover**, a real-time video object removal. In RT-Remover, we compose tracking and removal into a joint learning model, eliminating the overhead of an external tracking stage. Meanwhile, we employ an auto-regressive diffusion transformer that generates frames sequentially using a limited length key-value (KV) cache strategy, reducing memory and computation while maintaining long-term dependencies. In addition, we speed up the inference by distilling the auto-regressive diffusion model using distribution matching distillation and flow matching loss, and thus reduce the number of sampling steps from 25 to 2 while preserving background consistency.

**Our main contributions are summarized as follows:**

- We compose object tracking and removal into a single model that learns object tracking and object removal jointly, and thus remove the necessity for a separate tracking component. This integration significantly simplifies the overall efficiency of the video object removal pipeline.

- We develop an auto-regressive diffusion framework with efficient KV cache mechanism that balances temporal consistency and resource efficiency. Moreover, we perform the distribution matching distillation of auto-regressive remover and thus the model can inference with only 2 sampling steps. In addition, we train a faster VAE for our RT-Remover.

- Through extensive visualizations, quantitative comparisons, and ablation studies, we demonstrate the effectiveness of RT-Remover in real-time object removal, validating the impact of each proposed component. Finally, as shown in Figure 1 and Table 1, *RT-Remover reduces inference time and latency by a factor of 14 and 285, respectively, compared to Minimax-Remover.*

## 2 METHODOLOGY

RT-Remover has three major improvements compared with traditional remover, including composing object tracking and object removal into a joint learning process that avoids an individual object tracking, a new video object removal model based on auto-regressive diffusion model, and a lightweight VAE encoding/decoding. First, removing mask tracking brings in most improvement. Second, changing prediction method from diffusion model with full attention to auto-regressive diffusion model with unidirectional attention, further reduces the latency. Finally, distilling the original Wan2.1 VAE into a lightweight VAE further speeds up the inference.

### 2.1 ARCHITECTURE OVERVIEW

We adopt Wan2.1 (Wang et al., 2025) as our base model but remove the cross-attention module, following Minimax-Remover (Zi et al., 2025a), to enhance efficiency. Our model employs a 2-stage strategy. In Stage-1, we train an auto-regressive inpainter on a large-scale video dataset. In Stage-2,

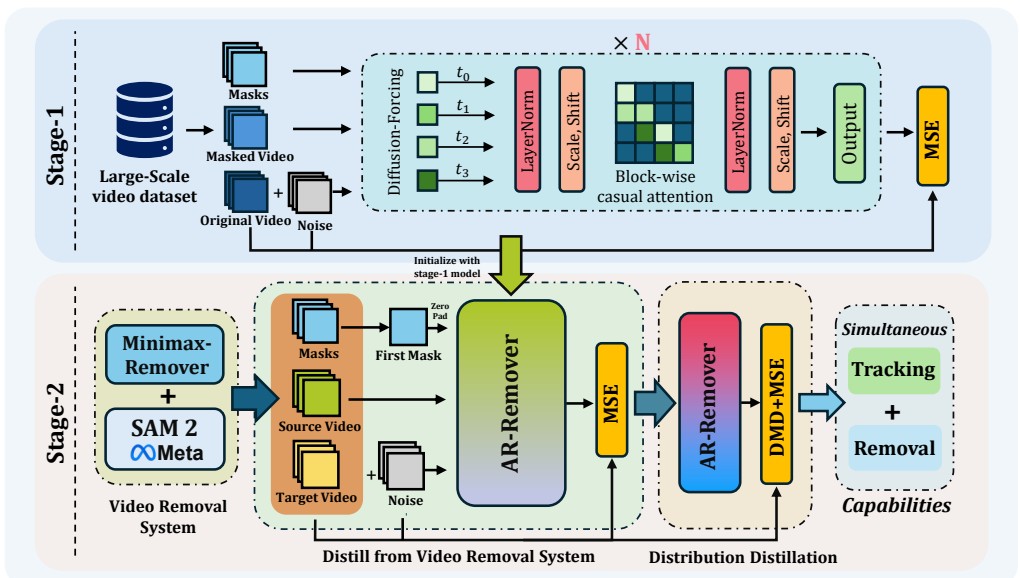

Figure 3: Our AR-Remover consists of two training stages. In Stage-1, we train an auto-regressive inpainter on a large-scale video dataset. In Stage-2, we use a general object remover model to distill our video object removal model that is initialized by the model from Stage-1. Two key differences in the inputs between models in Stage-1 and Stage-2 are the masked source video vs the source video, and the object masks for all frames vs the object mask for only the first frame. This distilled model after Stage-2 is capable of simultaneously tracking and removing objects in auto-regressive form.

we train video object removal based on the model obtained from Stage-1. We only use the initial mask of the first frame as our mask input. In summary, Stage-1 serves as a pretrained model, and Stage-2 is our final targeted video object removal model.

*Stage-1 Architecture.* Inputs are concatenated as $[\boldsymbol{x}_t, \boldsymbol{x}_m, \boldsymbol{m}]$, where $\boldsymbol{x}_t, \boldsymbol{x}_m \in \mathbb{R}^{b \times c \times f \times h \times w}$ denote noisy and masked latents, and $\boldsymbol{m}$ is the resized binary mask sequence. The initial mask $\boldsymbol{m}_0$ is duplicated across four channels, while each subsequent $\boldsymbol{m}_{1-N}$ encodes four reshaped masks from frames 1 to $4N + 1$. We use causal block-wise attention with limited KV length, which predicts the next frame auto-regressively, shown in Figure 3.

*Stage-2 Architecture.* Stage-2 adopts the same model architecture but modifies the input strategy. For the mask, only the first frame contains the resized initial mask, while subsequent frames are zero-padded. Instead of using the masked video, we directly input the source video. This design removes the need for an explicit object-tracking module, compelling the model to learn object removal and tracking jointly. At the same time, it allows the model to leverage Stage-1's inpainting capability in an auto-regressive manner, progressively achieving both tracking and object removal.

## 2.2 TRAINING PIPELINE

Our training pipeline consists of two successive stages. **Stage 1** adapts the Wan2.1 backbone to causal attention form with limited KV length. We remove the cross attention modules from the model for faster inference. **Stage 2** distills a general-purpose remover into a joint object tracking and video remover model.

### 2.2.1 STAGE 1: TRAINING AUTOREGRESSIVE FOUNDATION

In the first stage, we train an inpainter on the large-scale annotated dataset WebVid-10M (Bain et al., 2021) with uni-directional attention to establish the autoregressive foundation. This setup enforces an autoregressive learning scheme that maintains consistency between background and masked regions, ensures temporal coherence across frames.

Figure 4: The inference pipeline of our AR-Remover. Given the video and the object mask for only the first frame, the system auto-regressively generates removal results chunk-by-chunk in real time.

We adopt diffusion forcing to construct an autoregressive inpainter due to its efficiency and memory advantages. Specifically, it consumes only 1-4 of the GPU memory in attention compared to teacher forcing (Zhou et al., 2025), and unlike self-forcing, it does not rely on the generation of previous frames. During training, random masks are applied to the original videos to generate masked inputs, denoted as $[\boldsymbol{x}_t, \boldsymbol{x}_m, \boldsymbol{m}]$. These inputs are concatenated and fed into a DiT model. Each frame is assigned a distinct diffusion timestep by introducing different noise levels. Meanwhile, block-wise causal attention with row-wise sparsity ensures that each block can only attend to its preceding blocks. Our model is trained with a flow-matching objective using an MSE loss, where the target is defined as $\boldsymbol{\epsilon} - \boldsymbol{x}$, with $\boldsymbol{\epsilon}$ representing random noise and $\boldsymbol{x}$ denoting the latent representation of the target video. After training, the inpainter can autoregressively complete masked regions by leveraging the KV cache, thereby maintaining both spatial and temporal coherence.

### 2.2.2 STAGE 2: JOINT TRACKING AND REMOVAL

***Building Unified AR-Remover***. Inspired by the PropGen (Liu et al., 2024) and Senorita (Zi et al., 2025b), which only provides the first edited frame and then propagates the information to the whole video, we believe that the DiT model has strong ability of propagating information. It is natural to expand: the DiT can edit the video with only the information given in the first frame, no matter what types of this information. Moreover, multitask learning in one model is common in the computer vision field, thus we compose the tasks of object tracking and object removal into a single model. Therefore, we assume that only giving the first initial mask and the DiT model can removes marked object successfully. The input for the AR-Remover model is denoted as $[\boldsymbol{x}_t, \boldsymbol{x}_s, \bar{\boldsymbol{m}}]$, where $\boldsymbol{x}_t$ represents noisy latents obtained by adding noise to the target latents according to the noise scheduler, $\boldsymbol{x}_t$ refers to the source latents encoded from the source video, and $\bar{\boldsymbol{m}}$ is the mask condition.

*AR-Remover and the Stage-1 model have the following two distinct differences.*

- They have different mask inputs, $\bar{\boldsymbol{m}}$ vs $\boldsymbol{m}$. In AR-Remover, only the first frame contains the mask information, while all subsequent frames are zero-padded. In Stage-1 model, region masks in all frames need to be provided,

- They have different source inputs, $\boldsymbol{x}_s$ vs $\boldsymbol{x}_m$. In AR-Remover, the input $\boldsymbol{x}_s$ denotes the latent of the original source video, but in Stage-1 model, $\boldsymbol{x}_m$ denotes the latent of the masked source video.

In training, we use the same KV length as in Stage-1, but enforces the model can only access the left $N$ blocks, the training target is the flow matching target, using $\boldsymbol{v} = \boldsymbol{\epsilon} - \boldsymbol{x}_s$ as the target. After training, our remover learns to track and remove objects simultaneously in an auto-regressive form.

***Fast Distillation.*** The diffusion process typically requires a large number of iterative steps, resulting in substantial computational overhead and high latency in inference. Such inefficiency hinders the feasibility of real-time inference. To accelerate the process, we adopt the DMD2 framework (Yin et al., 2024) to distill the auto-regressive model. However, since the original distillation objective is tailored for video generation and neglects the static background in video object removal, we introduce a modified loss function:

$$\mathcal{L} = \text{DMD2}(f(\boldsymbol{x})) + \text{MSE}(f(\boldsymbol{x}), \boldsymbol{v}). \tag{1}$$

The MSE loss in this formulation explicitly requires to promise the reconstruction quality of the background while preserving the advantages of DMD2 distillation. After training, our approach reduces the number of inference steps from 50 to just 2 while still preserving a high quality of video object removal. Our distillation enables an efficient and real-time deployment.

***VAE Optimization.*** The original Wan2.1 VAE demonstrates strong performance in video reconstruction but suffers from slow inference, making it impractical for real-time setting. To address this limitation, we finetune LeanVAE (Cheng & Yuan, 2025) to align it with the latent space of Wan2.1 VAE. We term this new VAE as **Wan-LeanVAE**. Wan-LeanVAE achieves a significant speedup by a factor of 12 compared to the original Wan2.1 VAE, while preserves a comparable reconstruction quality as Wan2.1 VAE. Therefore, Wan-LeanVAE is more suitable for a real-time deployment.

After putting AR-Remover, fast distillation and Wan-LeanVAE togather, we reach a strong and real-time video object removal, termed as **RT-Remover.**

## 3 EXPERIMENTS

### 3.1 EXPERIMENTS SETUP

*Dataset.* We use the watermark-free WebVid-10M (Bain et al., 2021) and 400K videos from the Pexels(Pexels, 2024). We use CogVLM2 Hong et al. (2024) to get object names by asking the llm. We then use the grounded-sam2 (Liu et al., 2023; Ravi et al., 2024) to get the masks, where the groundingdino we use is the model with Swin Base backbone (Liu et al., 2021), while the SAM2 we choose is the hiera large version. We use the Scenedetect Castellano (2014) to filter out the video that has more than one scene, the threshold is 20.

*Training Details of Stage-1 Model.* We use AdamW (Loshchilov & Hutter, 2017) as the optimizer, with a learning rate of 1e-5 and a weight decay of 1e-6. The model architecture is same as the minimax-remover, the parameters are initialized with Wan2.1. The learning rate scheduler is set to constant. We use full fine-tuning, which means all parameters are trained. The batch size is set to 32, the resolution of video is $480 \times 832$, the frames are 81. Additionally, mixed precision and gradient checkpointing are employed to reduce GPU memory usage and accelerate the training process. We use 10 days to train the stage-1 model on 2 A800 nodes. The patch embedding layer is extended from 16 channels to 36 channels, the first 16 channels are initialized with Wan2.1 parameters and the rest 20 channels are initialized with zeros. We use flow matching(Lipman et al., 2023) scheduler to train, making DiT model to predict the velocity and setting shift with 3.

Table 2: Performance comparison of different methods on server with one 5090 GPU. To make fair comparison, we disable the external library, such as flash-attn. The best results are **boldfaced**.

| Method | Speed (FPS) | Diffusion Latency (s) | SAM2 Latency (s) | Overall Latency (s) |
|---|---|---|---|---|
| ProPainter | 2.25 | 16.85 | | 35.89 |
| COCOCO | 0.31 | 238.08 | | 257.12 |
| FloED | 0.64 | 107.17 | | 126.21 |
| Diffuseraser | 1.70 | 28.36 | 19.04 | 47.40 |
| VideoPainter | 0.12 | 617.6 | | 636.64 |
| VACE | 0.32 | 226.5 | | 245.54 |
| MiniMax-Remover | 2.36 | 15.16 | | 34.20 |
| RT-Remover (Ours) | **33.06** | **0.12** | **0** | **0.12** |

*Training Details of Stage-2 Model.* We initialized stage-2 model with the parameters of stage-1 model. The minimax-remover is used for dataset construction, with 12 inference steps, the resolution is $480 \times 832$ and 81 frames. We obtained 300K videos to train our stage-2 model using GPT-5 to filter the edited videos. We trained on these videos for 10 epochs and spent 2 days on 2 A800 nodes. The rest setting is same as the stage-1 model.

*The Training Details of Fast Distillation.* We use DMD2 (Yin et al., 2024) to distill our model. The learning rate is 1e-5, the training steps is 6000, the batch size is 32. The rest experiment details are same as the *training details of stage-2 model.*

*Inference Details of AR-Remover.* We conduct inference on a 5090 GPU. The inference process is configured with two steps, a resolution of $480 \times 832$, and a frame length of 81. Our model does not employ classifier-free guidance. Under this setting, the model achieves a GPU memory usage

Table 3: Comparison of different methods on the DAVIS and Pexels datasets. The best results are highlighted in **bold**. "TC" denotes temporal consistency, "VQ" stands for visual quality, and "Succ" represents the success rate. We implemented baselines for the video object removal task.

| Method | DAVIS Dataset | | | | | Pexels Dataset | | | | |
| | Quantitative Results | | | GPT-5 Eval | | Quantitative Results | | | GPT-5 Eval | |
| | SSIM | PSNR | TC | VQ | Succ | SSIM | PSNR | TC | VQ | Succ |
|---|---|---|---|---|---|---|---|---|---|---|
| Diffusion-forcing | 0.9567 | 33.27 | 0.9636 | 4.54 | 46.67% | 0.9824 | 36.34 | 0.9920 | 3.91 | 23.50% |
| Self-Forcing | 0.9532 | 33.35 | 0.9621 | 5.25 | 57.78% | 0.9812 | 36.22 | 0.9910 | 5.16 | 52.50% |
| Teacher-Forcing | 0.9297 | 32.69 | 0.9596 | 4.94 | 54.44% | 0.9687 | 35.42 | 0.9885 | 4.41 | 36.00% |
| Ours | **0.9688** | **33.62** | **0.9685** | **5.70** | **65.56%** | **0.9846** | **37.02** | **0.9924** | **6.12** | **72.50%** |

of **12.4 GB**, runs at **33 FPS**, and maintains a latency of **0.12 s**. For evaluation, we adopt the DAVIS dataset, which contains 90 videos, and a set of 200 videos from Pexels.

## 3.2 EXPERIMENTAL RESULTS

***Quantitative Results***. As shown in Table 3, our method consistently outperforms all baselines on both the DAVIS(Pont-Tuset et al., 2017) and Pexels datasets(Pexels, 2024). On DAVIS, we achieve the highest SSIM of 0.9688, PSNR of 33.62, and temporal consistency of 0.9685. These results indicate that our approach preserves fine-grained structures, produces sharper reconstructions of background, and maintains

Table 4: User study. The best results are in **bold**. D-Forcing, T-Forcing and S-Foricing mean Diffusion-Foricing, Teacher-Forcing and Self-Forcing.

| D-Forcing | T-Forcing | S-Forcing | Ours |
|---|---|---|---|
| 22.05% | 38.97% | 30.76% | **82.05%** |

stable temporal dynamics better than competing methods. Similarly, on the Pexels dataset, our model reaches an SSIM of 0.9846, PSNR of 37.02, and temporal consistency of 0.9924, all surpassing prior approaches by a clear margin. Interestingly, Diffusion-forcing has higher background and temporal consistency than teacher-forcing and self-forcing on quantitative results, which gives the rationality to use it as the first-stage pretraining method. These consistent gains across datasets demonstrate both the accuracy and robustness of our method.

***Qualitative Results***. Beyond quantitative measurements, our method also shows clear advantages in qualitative results. The GPT-5 based analysis rates our approach the highest in terms of visual quality and removal success. On DAVIS, our method attains a score of 5.70 with a success rate of 65.56%, outperforming all baselines. On Pexels, the improvements are even more pronounced, with a visual quality score of 6.12 and success rate of 72.50%, higher than the second best performance with the score of 5.16 and 52.50% (i.e., self-forcing). User studies further confirm these findings, showing a strong preference for our outputs compared with CausVid(Yin et al., 2025), self-forcing(Huang et al., 2025), and teacher-forcing(Zhou et al., 2025) methods. We also conduct a user study in which users are asked to select their preferred options from a set of given videos. As shown in Table 4, the results demonstrate that our method achieves the best performance according to human evaluations.

## 3.3 ABLATION STUDY

We conduct ablation studies to evaluate the contributions of different design choices, focusing on two aspects: (1) the impact of pretraining weights, and (2) the effect of window size.

**Comparing different pretrained weights.** Table 6 compares the performance of several pretraining models. Wan2.1 and MiniMax-Remover weights produce reasonable results, but the proposed Stage-1 model consis-

Table 5: Comparison of different VAEs. The evaluation resolution is $480 \times 832$ with 81 frames. The speed was tested on 5090 GPUs. The best results are in **bold**.

| VAE | FPS | SSIM | PSNR | L1 $\times e^{-2}$ | L2 $\times e^{-3}$ |
|---|---|---|---|---|---|
| Wan2.1 | 16.03 | **0.9786** | **38.75** | **2.00** | **1.97** |
| TAEHV | **343.2** | 0.9655 | 36.89 | 2.75 | 3.62 |
| Wan-LeanVAE (ours) | 206.3 | 0.9743 | 38.13 | 2.17 | 2.40 |

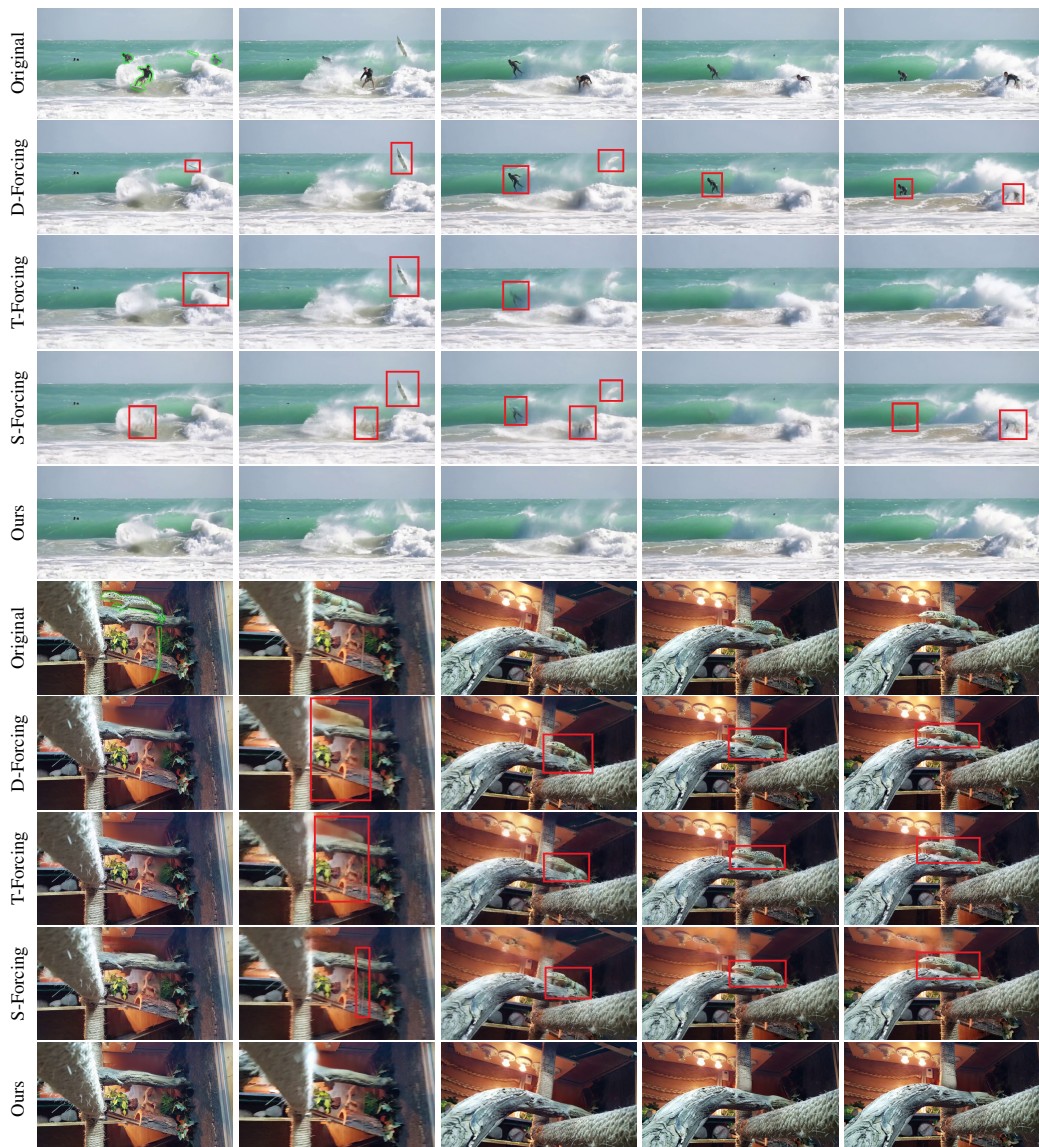

Figure 5: Qualitative comparison of different methods. For the baselines and our remover, only the mask for the first frame is given. D-Forcing, T-Forcing and S-Foricing donate Diffusion-Foricing, Teacher-Forcing and Self-Forcing. Undesired objects or artifacts are highlighted with red rectangles.

tently delivers superior performance. With the support of a causal attention mechanism, Stage-1 achieves the highest quantitative scores. It also receives the best evaluations in GPT-5 Eval with visual quality 5.70 and success rate 65.56%. These results show that Stage-1 is closer to the final weights and leads to significantly improved task-specific performance.

Table 6: Ablation study. We compare different pretraining weights and prove the effectiveness of The Stage-1 model. The best results are in **bold**.

| Method | Pretraining Model | Task | Attention | Quantitative Results | | | GPT-5 Eval | |
|--------|-------------------|------|-----------|------|------|------|------|------|
| | | | | SSIM | PSNR | TC | VQ | Succ |
| Ab-1 | Wan2.1 | Gen | Bidirectional | 0.9522 | 33.6 | 0.9610 | 3.58 | 27.78% |
| Ab-2 | MiniMax-Remover | Remove | Bidirectional | 0.9539 | 33.52 | 0.9655 | 4.97 | 57.78% |
| Ab-3 | Stage-1 Model | Remove | Unidirectional | **0.9688** | **33.62** | **0.9685** | **5.70** | **65.56%** |

Table 7: Ablation study of different window size. The speed and GPU memory consumption was tested on 5090 GPUs. "All" means window size is infinite. The best results are in **bold**.

| Method | Window Size | Performance | | | Quantitative Results | | | GPT-5 Eval | |
|--------|-------------|-------------|------|---------|--------|--------|--------|------|---------|
| | | Memory | FPS | Latency | SSIM | PSNR | TC | VQ | Succ |
| Ab-1 | 3 | **11.7GB** | **35.84** | **0.1076s** | 0.9621 | 34.04 | 0.9685 | 5.10 | 55.56% |
| Ab-2 | 5 | 12.4GB | 33.06 | 0.1200s | **0.9688** | 33.62 | 0.9685 | 5.70 | 65.56% |
| Ab-3 | 9 | 13.5GB | 28.52 | 0.1432s | 0.9621 | 33.21 | **0.9695** | **5.82** | 67.78% |
| Ab-4 | 8 | 18.5GB | 25.47 | 0.2113s | 0.9629 | **33.90** | 0.9687 | 5.80 | **68.89%** |

**Comparing different VAEs**. We train Lean-VAE (Cheng & Yuan, 2025) to align its latent space with that of Wan2.1. The experimental results are presented in Table 5. We can see that Wan-LeanVAE obtains comparable performance with Wan2.1, while achieves a much higher performance than taehv (Boer Bohan, 2025).

**Ablation study of different window size.** Table 7 analyzes the trade-offs introduced by different window sizes. A small window such as size 3 offers the best efficiency in terms of memory usage, with 11.7GB, and speed, reaching 35.84 FPS, but it shows a slight drop in quality. Increasing the window size to 5 or 9 provides a better balance, with size 9 achieving the highest temporal consistency and visual quality. Using a full window, referred as All, further improves the successful rate to 68.89%, but requires significantly more memory at 18.5GB and results in slower inference speed of 25.47 FPS. These results indicate that practical deployment demands a careful trade-off between accuracy and efficiency. Therefore, we choose window size as 5 in our experiment to balance the speed and performance.

**Visualization of Attention Map.** To justify the choice of a window size of 5, we visualize the attention maps. Specifically, we sample 100 videos from the Pexels dataset and evaluate them using a bidirectional model where only the first mask is provided. We then compute the average attention map across 30 layers at different timesteps. The result is shown in Figure 6. We cna see that queries tend to attend to nearby keys more strongly.

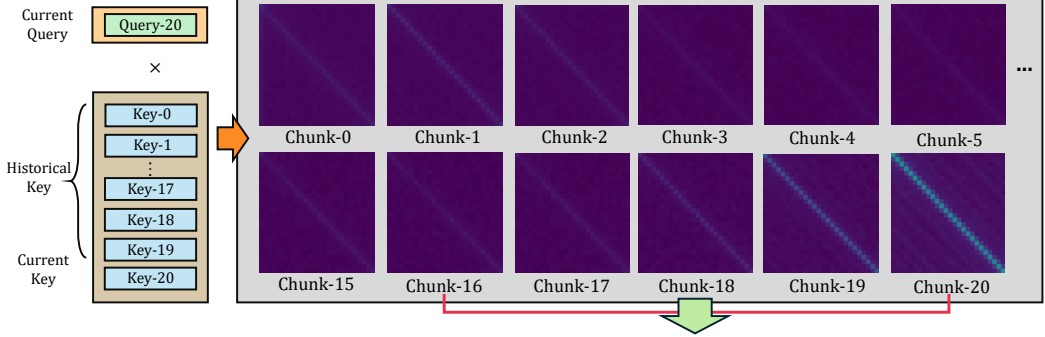

Figure 6: Visualization of the Attention Map in the Bidirectional-Attention Video Object Remover. We use a video consisting of 81 frames, which is encoded into 21 latent chunks. We can see that queries tend to attend to nearby keys more strongly.

## 4 CONCLUSION

In conclusion, we presented RT-Remover, a novel real-time video object removal that achieves an unprecedented low latency. By composing object tracking and inpainting into a joint process, employing an auto-regressive diffusion model with a caching mechanism, and leveraging model distillation to drastically reduce sampling steps, our approach simplifies the pipeline and enables real-time inference. Extensive experiments demonstrate that RT-Remover achieves state-of-the-art latency of 0.12s and a high inference of 33 FPS, while maintaining competitive visual quality, making it a practical solution for real-time interactive video editing.

**Ethics statement** This work adheres to the ICLR Code of Ethics. Our research does not involve human subjects, studies with potential for harm, or methodologies raising concerns regarding discrimination, bias, fairness, privacy, or security. No human-annotated datasets were used in the process; all data processing and model training rely on publicly available or synthetically generated resources in compliance with legal and ethical standards. We have ensured research integrity through rigorous documentation and reproducibility efforts, as detailed in the Reproducibility Statement.

**Reproducibility Statement** To facilitate reproducibility of our results and to ensure that our methodology can be reliably adopted in future research, we provide extensive details on the training parameters, including hyperparameter configurations, optimization strategies, and model initialization procedures, which are systematically reported in the Experiments section.

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

## A  LLM USAGE

In preparing this manuscript, we used ChatGPT solely for language polishing and minor refinements aimed at improving clarity, grammar, and overall flow. Sections of the draft were provided to the LLM for suggested revisions, which were subsequently reviewed, edited, and incorporated by the authors where appropriate. All core ideas, research contributions, technical details, and analyses are entirely the work of the authors and were not generated or conceived by the LLM. No other large language models were utilized during the research process.

## B  DATASET INTRODUCTION

Our dataset consists of two main sources: the watermark-free Webvid-10M (Bain et al., 2021) and 400K videos collected from Pexels.com (Pexels, 2024). We annotate these videos using a combination of LLM-based labeling and Grounded-SAM2. Specifically, we employ CogVLM2 (Hong et al., 2024) to extract object categories from each video, and then apply Grounded-SAM2 (Liu et al., 2023; Ravi et al., 2024) to detect and track the corresponding objects throughout the entire sequence. To ensure consistency, we further filter out videos containing multiple scenes using PySceneDetect (Castellano, 2014). In the stage-1 training, we only use the WebVid-10M videos. In the second stage of training, we leverage Minimax-Remover to construct video triplets with Pexels videos and filtered by GPT-5 (OpenAI, 2025), each consisting of a source video, an edited video, and the associated masks. This process yields a total of 300K high-quality video triplets, which serve as the foundation for robust model training.

## C  RELATED WORKS

### C.1  VIDEO INPAINTING

Video inpainting methods can generally be divided into two categories: **text-guided** and **non-text-guided** approaches. The former completes masked regions based on textual prompts, while the latter focuses on removing objects or repairing corrupted areas without relying on text.

In recent years, a number of **text-guided video inpainting** methods Wang et al. (2024); Zhang et al. (2023b); Zi et al. (2024); Bian et al. (2025); Yang et al. (2025); Zi et al. (2025a;b); Hu et al. (2024); Jiang et al. (2025) have emerged. Early work such as VideoComposer (Wang et al., 2024) demonstrated the feasibility of combining multiple input conditions to guide inpainting with text. Building upon this idea, AVID incorporated image inpainting models and introduced a motion layer trained to propagate edits across frames. COCOCO (Zi et al., 2024) further enhanced this approach by adding damped global attention and textual cross-attention, significantly improving temporal consistency and user control. It also introduced a mechanism for personalized video inpainting. Meanwhile, VIVID (Hu et al., 2024) contributed a large-scale dataset containing 10 million image and video pairs for localized editing, which enabled the training of a powerful text-guided inpainting model. Expanding the application scope, MTV-Inpaint (Yang et al., 2025) unified both conventional scene completion and novel object insertion within a single framework. Additionally, VideoPainter (Bian et al., 2025) leveraged a DiT-based architecture with an efficient context encoder to handle masked inputs and inject background priors into a pre-trained video DiT, allowing plug-and-play inpainting. VACE (Jiang et al., 2025) pushed this further by integrating an inpainting model with Control-Net (Zhang et al., 2023a) to support diverse editing tasks, achieving state-of-the-art results. Lastly, Senorita-Remover (Zi et al., 2025b), designed for the Senorita-2M dataset, adopted a contrastive prompt strategy to specialize in object removal from videos.

On the other hand, **non-text-guided video inpainting**, also known as video object removal, typically avoids the use of text prompts and focuses instead on direct content manipulation.(Zhou et al., 2023; Lee et al., 2025; Gu et al., 2024; Li et al., 2025; Zi et al., 2025a) For example, ProPainter(Zhou et al., 2023) was an early method that first completed optical flow and then used a vision transformer to fill masked regions. Following this, FFF-VDI(Lee et al., 2025) introduced a diffusion-based approach that propagates noise latents from future frames into masked regions and fine-tunes a pre-trained image-to-video diffusion model for final synthesis. FloEDGu et al. (2024) combined optical flow and optional textual embeddings, injecting both into the inpainting pipeline to enhance object

removal. Building upon existing work, DiffuEraser(Li et al., 2025) utilized ProPainter to generate initial removal results, which were then inverted into latents and reconstructed using a specialized video inpainting model. More recently, MiniMax-Remover(Zi et al., 2025a) introduced a robust training strategy using minimax optimization: the inner loop searches for adversarial noise that hinders inpainting performance, while the outer loop trains the model to succeed even under such challenging conditions.

## C.2 STREAMING VIDEO GENERATION

Generating video auto-regressively become more and more popular in recent days. Recent advances in video generation have primarily followed two distinct paradigms: language model-based token prediction and diffusion-based synthesis. We review representative works from both lines of research, with a focus on their approaches to temporal modeling and efficiency.

**LLM-Based Video Generation**. VideoPoet (Kondratyuk et al., 2023) demonstrates a unified generative framework that models video, image, audio, and text as token sequences. It leverages a visual tokenizer (i.e., MagViT-v2 (Yu et al., 2023)) to convert video frames into discrete tokens, and uses a large language model to autoregressively predict the next token in the sequence. This approach benefits from the scalability and long-range modeling capabilities of LLMs, enabling coherent video generation over extended durations.

**Diffusion-Based Video Generation**. Diffusion models have become the dominant approach for high-quality video synthesis, owing to their strong generative fidelity. However, generating long videos with standard per-frame diffusion is computationally intensive. Recent work has proposed more efficient variants based on chunk-wise prediction (Yin et al., 2025; Huang et al., 2025; Chen et al., 2025; Sand.ai et al., 2025; Deng et al., 2024; Jin et al., 2025).

CausVid (Yin et al., 2025) adapts a bidirectional DiT into an autoregressive transformer that generates frames in a causal order. It introduces a novel distillation framework, reducing a 50-step diffusion process into 4 steps by initializing the student from the teacher's ODE trajectories and applying asymmetric supervision. SkyReels-V2 (Chen et al., 2025) proposes a diffusion forcing strategy with a non-decreasing noise schedule to restrict the generation search space. This approach enables efficient long-range video synthesis with stable temporal dynamics. MAGI-1(Sand.ai et al., 2025) models the denoising process at the chunk level, where noise levels increase monotonically over time. This design facilitates causal temporal modeling and supports streaming generation with consistent frame quality. NOVA (Deng et al., 2024) preserves causal modeling across frames while introducing bidirectional attention within individual frames. This hybrid design improves generation quality without compromising temporal consistency. PyramidFlow (Jin et al., 2025) adopts a hierarchical strategy, where each stage predicts the next frame conditioned on prior stages. This promotes a coarse-to-fine generation process, balancing quality and efficiency. Self-Forcing (Huang et al., 2025) introduces autoregressive rollout during training, where each frame is conditioned on self-generated outputs. It combines a few-step diffusion model with stochastic gradient truncation and a rolling key-value cache mechanism, significantly improving autoregressive video extrapolation efficiency.

**Remark.** Most of previous video object removal methods rely on object tracking, as a preprocessing step, to generate mask sequences across all frames. This process significantly hampers inference speed and remains a major bottleneck for real-world applications. Meanwhile, these models operate in a non-auto-regressive form, and thus increases latency and diminishes interactivity. In RT-Remover, we compose the object tracking and video object removal into a single auto-regressive diffusion model. Meanwhile, we also largely reduce the sampling steps in inference from 25 to 2 by a model distillation and speed up the VAE encoder and decoder via training a lightweight and well-performing VAE. Our model is the first real-time video object removal.

## D WHY WE CAN TRUST GPT-5 FOR EVALUATION?

Here, we report the evaluation results of GPT-5 on a benchmark of 100 video object removal samples, consisting of an equal split between success cases (50%) and failure cases (50%). As shown in Table 8, GPT-5 demonstrates the strongest performance among all evaluated large language models, achieving 97% agreement with human judgments. Notably, GPT-5 exhibits both high accuracy in

Table 8: LLM performance on evaluating video object removal.

| Success Rate | GPT-4O | GPT-4O-mini | GPT-4-turbo | GPT-O3 | GPT-5 |
|---|---|---|---|---|---|
| Acc in Failure Cases | 94.0% | 98.0% | 86.0% | 96.0% | 98.0% |
| Acc in Success Cases | 88.0% | 26.0% | 58.0% | 94.0% | 96.0% |
| Overall Accuracy | 91.0% | 62.0% | 72.0% | 95.0% | 97.0% |

distinguishing successful removals and strong reliability in identifying failure cases, highlighting its robustness and consistency in assessing video object removal quality.

# E    MORE QUALITATIVE RESULTS OF OUR RT-REMOVER

More qualitative results can be seen in Figure 7.

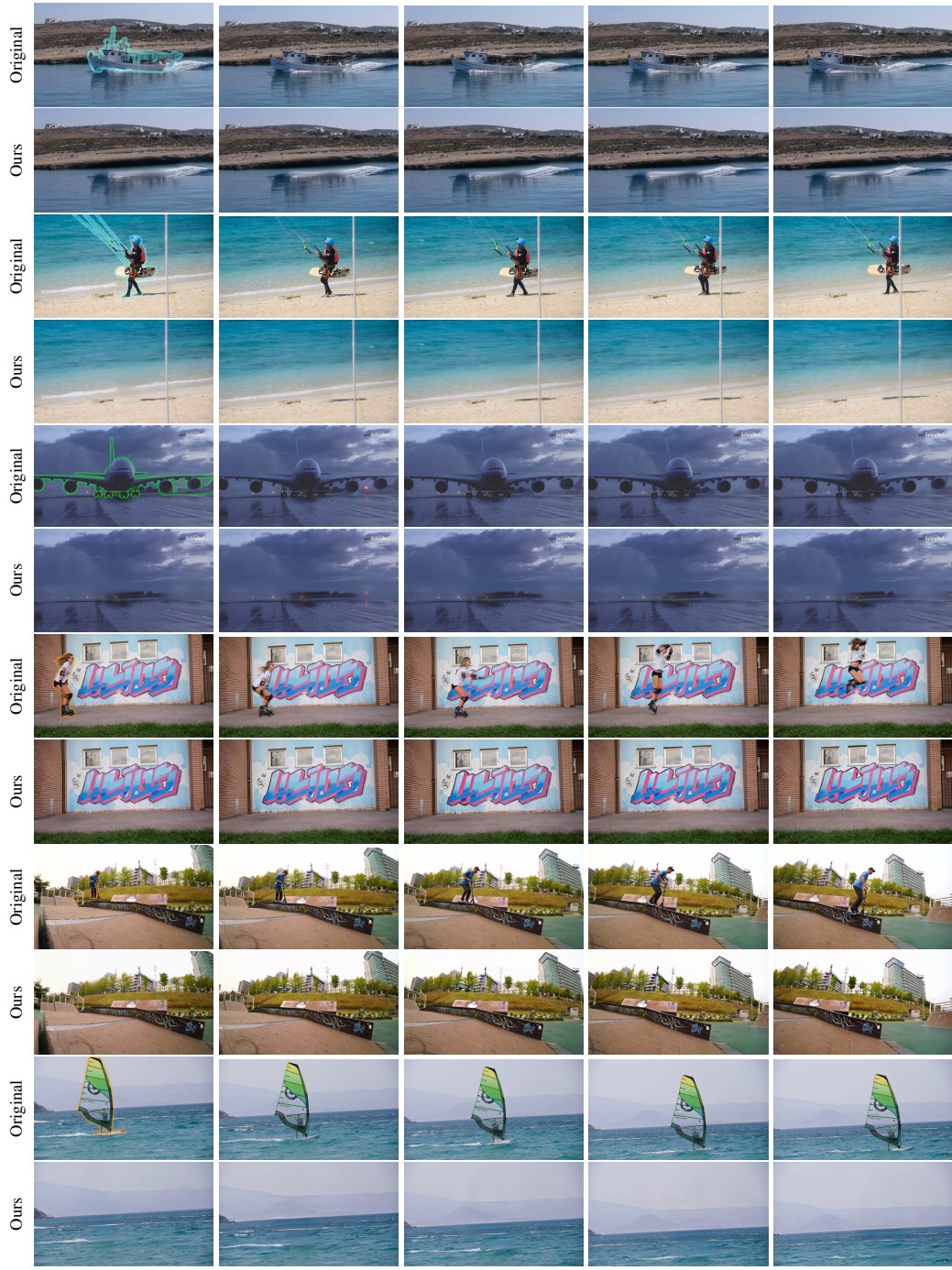

Figure 7: More visual results of our RT-Remover.

