# OpenReview forum: "RT-Remover: A Real-Time Video Object Removal by Composing Tracking and Removal in Auto-Regressive Diffusion Transformers"
_ICLR.cc/2026/Conference — Submitted to ICLR 2026_

### Official Review · Reviewer_AA5H · 2025-10-29

**Soundness:** 3
**Presentation:** 3
**Contribution:** 3
**Rating:** 4
**Confidence:** 3

**Summary:**

This paper presents RT-Remover, a real-time video object removal system that merges tracking and inpainting into a single auto-regressive diffusion model. It requires only a starting mask for the first frame and uses causal attention with a key-value cache for efficient sequential generation. Through step distillation (distribution matching distillation) and a lightweight VAE, the method reduces sampling steps to two, achieving 33 FPS and 0.12s latency while maintaining strong visual quality and temporal consistency.

**Strengths:**

1) The paper introduces a single model that jointly performs tracking and inpainting, removing the need for separate stages and simplifying the pipeline significantly.
2) By combining auto-regressive diffusion with key-value caching and applying a tailored distillation strategy, the method reduces sampling steps from 25 to 2 while maintaining quality.
3) The approach achieves 33 FPS and 0.12s latency, making real-time interactive video editing feasible.

**Weaknesses:**

1) Lack of qualitative results: it would be great to see the examples of edited videos (not rolled out frames) to assess the quality of removal by the model (especially temporal consistency).
2) Lack of qualitative and quantitative comparisons: in Table 3 and Table 4, why not compare to Minimax-Remover?
3) Minor grammatical errors: for instance, in line 138 "simplies" -> "simplifies", in lines 280-281 "togather" -> "together", and so on.

**Questions:**

1) How does the model handle inaccurate initial masks or cases with multiple objects? Is there any quantitative or qualitative analysis on robustness?
2) Can the authors share video examples of the removal results? Static images are insufficient to judge temporal consistency and overall quality.
3) Table 7 evaluates fixed window sizes, but have you considered adaptive strategies where the KV cache length changes based on motion or chunk complexity? Could this further optimize efficiency while preserving quality?

---

### Official Review · Reviewer_NmWE · 2025-10-30

**Soundness:** 2
**Presentation:** 2
**Contribution:** 3
**Rating:** 4
**Confidence:** 4

**Summary:**

This paper introduces RT-Remover, a lightweight, low-latency, and autoregressive diffusion-based video object removal model. A key feature of RT-Remover is its ability to remove masked objects from an entire video by only requiring a mask for the first frame.

The authors propose a two-stage training strategy to realize this model:
1. Stage 1: A pre-trained Wan2.1 generative model is fine-tuned into an auto-regressive general inpainting model.
2. Stage 2: The model undergoes distillation using a distribution matching distillation method. This process leverages object removal data generated by Minimax-Remover to transform the model into a lightweight object removal solution that requires only 2 inference steps and can simultaneously track and remove objects based solely on the first-frame mask.

Furthermore, the authors replace the VAE with LeanVAE to achieve further acceleration.

**Strengths:**

1. This is the first real-time video object removal model.
2. The proposed RT-Remover in this paper achieves extremely fast inference speed, which is 14× faster than state-of-the-art models.
3. The authors integrate both mask tracking and object removal functionalities into a single model, eliminating the need for an additional model to handle mask tracking. This not only enhances user-friendliness but also reduces the model latency.

**Weaknesses:**

1. Table 2 is not referenced or discussed anywhere in the main body of the paper. Additionally, the 'SAM2 Latency (s)' column in Table 2 appears incomplete, with several data missing.
2. The quantitative and qualitative comparative baselines are all generative models, but it’s unclear if they were trained on object removal datasets, making the fairness of comparison questionable. Additionally, no metrics are compared with mainstream video object removal models. To ensure fair comparison, these mainstream non-autoregressive models could be tested under an autoregressive inference setup (i.e., only inputting causal temporal frames) to align with RT-Remover’s inference logic.
3. While RT-Remover is designed to track and remove objects using only the first-frame mask, this input setting is unfair when comparing with other video object removal models, as those baselines lack the design to track objects across frames with just the first-frame mask.
4. Visualized results are limited, with no demos on challenging cases (e.g., target exiting and re-entering the frame). Such edge-case results would better validate robustness.

**Questions:**

See weaknesses.

---

### Official Review · Reviewer_WcC6 · 2025-10-31

**Soundness:** 2
**Presentation:** 2
**Contribution:** 2
**Rating:** 4
**Confidence:** 4

**Summary:**

RT-Remover is a real-time video object removal system that unifies object tracking and inpainting into a single streamlined process. It employs an auto-regressive diffusion model with distribution-matching distillation to reduce sampling steps from 25 to 2, achieving 0.12s latency and 33 FPS on a 5090 GPU. This approach significantly simplifies the pipeline while maintaining competitive visual quality and achieving the lowest latency among existing methods.

**Strengths:**

[1] The paper addresses an important problem, as video editing speed is a critical factor for enabling deployment on mobile platforms.

[2] The presented experimental results demonstrate excellent latency and efficiency.

**Weaknesses:**

[1] The paper reads as if various existing methods were simply combined for faster performance, giving the impression of a technical report describing a series of empirical design choices rather than a research paper presenting novel insights.

[2] What exactly is the problem being addressed, and what are the underlying causes of this problem?

[3] What is the proposed contribution for fast distillation? Is the main contribution merely the adoption of DMD2, or is there an additional methodological innovation?

[4] The current writing structure mostly follows a pattern of “problem → apply existing method”, which raises the question of whether this work truly qualifies as a research paper rather than an implementation summary.

[5] Is there any video results?

**Questions:**

My questions are in the weakness

---

### Official Review · Reviewer_SNRj · 2025-11-04

**Soundness:** 3
**Presentation:** 2
**Contribution:** 2
**Rating:** 6
**Confidence:** 4

**Summary:**

This paper proposes a real-time approach for video object removal by introducing three key innovations:
Joint tracking and inpainting: The method integrates object tracking and video inpainting into a unified process to improve temporal consistency and efficiency.
Auto-regressive diffusion with distillation: It leverages an auto-regressive diffusion model enhanced through a distillation technique to achieve high-quality, temporally coherent results in real time.
Fixed-length key-value cache: A fixed-length cache mechanism is employed to manage memory and computation effectively, enabling fast inference across video frames.

**Strengths:**

Belows are strong points that this paper has:

**1. Comprehensive and well-engineered approach:**
The paper demonstrates remarkable research and engineering effort toward achieving real-time video object removal. The authors systematically present a complete methodology that focuses on optimizing both efficiency and speed in the training and inference pipelines.

**2. Extensive experimental validation:**
The proposed method is thoroughly evaluated through a wide range of experiments, comparing both performance and efficiency against existing approaches.

**3. Multi-perspective evaluation:**
The paper provides convincing evidence of the method’s effectiveness through diverse evaluation metrics—including efficiency benchmarks, quantitative performance measures, GPT-5 assessments, and user studies, offering a well-rounded understanding of RT-Remover’s strengths.

**Weaknesses:**

Belows are weak points that this paper has:

**1. Incomplete performance comparison:**
The paper lacks a comprehensive experiment table comparing the proposed RT-Remover with other video object removal models in terms of model performance. Table 2 would be more informative if it combined both efficiency and performance metrics to provide a unified view of trade-offs.

**2. Insufficient methodological details:**
- The process of fine-tuning LeanVAE to align with the latent space of Wan2.1 VAE is not clearly explained and should be made self-contained for reproducibility.
- The fixed-length key-value cache mechanism and its impact on performance are not sufficiently detailed; currently, the only reference is Figure 6, which lacks quantitative or descriptive depth.
- The notation $N$ mentioned around Lines 194–195 is undefined or ambiguous and should be clarified in the text.

**3. Lack of failure case analysis:**
It would be valuable for the authors to include examples of failure cases. For instance, scenarios where the target object disappears and reappears within the video could help illustrate the model’s limitations and potential areas for improvement.

**Questions:**

Please check above the questions listed in Weaknesses section.

---

### Meta-Review · Area_Chair_MJ8C · 2025-12-30

**Summary:**

The paper received an average rating of 4.50 (4, 4, 4, 6). Reviewers raised concerns regarding the clarity of the paper’s contribution (WcC6), limited novelty beyond existing methods (WcC6), and incomplete experimental evaluation (SNRj, WcC6, NmWE, AA5H), including limited qualitative results and missing comparisons. As no rebuttal was submitted by the authors, these concerns remain unaddressed. The AC therefore recommends Reject.

**Reviewer Concerns:**

No rebuttal was provided by the authors. None of the reviewer concerns regarding the clarity of the paper’s contribution, limited novelty beyond existing methods, and incomplete experimental evaluation were addressed during the discussion phase, and all major issues raised by the reviewers remain outstanding.

**Reviewer Scores:**

As no rebuttal was provided, there was no additional information or evidence to support reconsideration of the evaluations, and all reviewers would be expected to maintain their initial ratings (4, 4, 4, 6).

---

### Decision · Program_Chairs · 2026-01-26

Reject